# NMR-Based Metabolomics Profiling for Radical Scavenging and Anti-Aging Properties of Selected Herbs

**DOI:** 10.3390/molecules24173208

**Published:** 2019-09-03

**Authors:** Mahanom Hussin, Azizah Abdul Hamid, Faridah Abas, Nurul Shazini Ramli, Ahmad Haniff Jaafar, Suri Roowi, Nordiana Abdul Majid, Mohd Sabri Pak Dek

**Affiliations:** 1Department of Food Science, Faculty of Food Science and Technology, Universiti Putra Malaysia, Serdang 43400, Selangor, Malaysia; 2Food Science and Technology Research Centre, Malaysian Agricultural Research and Development Institute (MARDI), MARDI Headquarters, Persiaran MARDI-UPM, Serdang 43400, Selangor, Malaysia; 3Laboratory of Natural Products, Institute of Bioscience, Universiti Putra Malaysia, Serdang 43400, Selangor, Malaysia

**Keywords:** radical scavenging, elastase inhibitor, collagenase inhibitor, metabolomics, multivariate data analysis, proton NMR, compound profiling

## Abstract

Herbs that are usually recognized as medicinal plants are well known for their therapeutic effects and are traditionally used to treat numerous diseases, including aging. This study aimed to evaluate the metabolite variations among six selected herbs namely *Curcurma longa*, *Oenanthe javanica*, *Vitex negundo*, *Pluchea indica*, *Cosmos caudatus* and *Persicaria minus* using proton nuclear magnetic resonance (^1^H-NMR) coupled with multivariate data analysis (MVDA). The free radical scavenging activity of the extract was measured by 2,2-diphenyl-1-picrylhydrazyl (DPPH), 2,2-azinobis(3-ethyl-benzothiazoline-6-sulfonic acid) (ABTS) and oxygen radical absorbance capacity (ORAC) assay. The anti-aging property was characterized by anti-elastase and anti-collagenase inhibitory activities. The results revealed that *P. minus* showed the highest radical scavenging activities and anti-aging properties. The partial least squares (PLS) biplot indicated the presence of potent metabolites in *P. minus* such as quercetin, quercetin-3-*O*-rhamnoside (quercitrin), myricetin derivatives, catechin, isorhamnetin, astragalin and apigenin. It can be concluded that *P. minus* can be considered as a potential source for an anti-aging ingredient and also a good free radical eradicator. Therefore, *P. minus* could be used in future development in anti-aging researches and medicinal ingredient preparations.

## 1. Introduction

Aging is characterized by a progressive decline in physiological functions of various organs, followed by dysfunctions, and ultimately death [1]. It can be categorized by physiological, metabolic, and immune disturbances which are related to extreme oxidative stress. Continual production of reactive oxygen species (ROS), although essential for biological functions, is deadly when present in excessive amounts. These radicals are ordinarily eliminated through the antioxidant defense system. However, at the aging stage, the antioxidant defense system becomes dysfunctional, resulting in cellulars damage and consequently rendering the cells vulnerable to various degenerative diseases.

Across the world nowadays, there is a diversity of herbs that have long been used traditionally by folk for aging either to be included in cooked meals or eaten raw as salad. Among the common herbs used in Malaysia for these purposes are the leaves of *Curcurma longa, Oenanthe javanica*, *Vitex negundo*, *Pluchea indica*, *Cosmos caudatus* and *Persicaria minus*. In addition to these purposes, these herbs are believed to have medical values that indirectly contribute to the health benefits of their consumers. For example, the leaves of *C. longa* are widely used due to their pleasant aroma, especially in Asian culinary preparations, although their use in medicinal preparation is not yet well documented. *O. javanica* is usually used in traditional medicine to treat jaundice, hypertension, polydipsia, lowering blood pressure, relieving arrhythmia and anti-anaphylaxis [2,3,4]. It was also reported that this herb has protective properties against oxidative stress-induced liver damage, hepatic lipid peroxidation in bromobenzene-treated rats, alcohol intoxication and hepatitis B virus activity [5,6,7,8]. *V. negundo* is traditionally used as folk medicine to cure headache, fever, cold and cough, treatment of eye-disease, toothache, inflammation, leukoderma, skin-ulcers, catarrhal fever, rheumatoid arthritis, gonorrhea, bronchitis, vermifuge, antibacterial, antipyretic and antihistaminic properties [9,10]. This herb has been reported to have anti-hyperglycaemic potential, hepatoprotective effects and has been used as hormone replacement therapy in clinical practice [11,12,13]. *P. indica* is known traditionally to possess antioxidant, antimicrobial, anti-inflammatory, exert hypoglycaemic and diuretic effects; and to cure rheumatoid arthritis [14,15,16,17,18,19]. A popular local herb, *C. caudatus* has been used to improve blood circulation, promote the formation of healthy bones, reduce body heat, promote fresh breath, treat infections associated with pathogenic microorganisms, lower high blood pressure and it is also useful in cleansing the blood [20,21,22,23,24]. This herb is also known to have antioxidant, anti-diabetic, anti-hypertensive, anti-bacteria and anti-fungal properties [22,25,26,27]. A well-known herb, *P. minus* has been reported to possess antimicrobial activity, cytotoxic activity against HeLa (human cervical carcinoma), a potent antioxidant and anticancer activity [28,29,30,31,32,33]. This herb is also used in herbal medicine to cure digestive disorders, as its essential oil has been used to remove dandruff and it is also used in aroma therapy and in the perfume industry [34,35]. However, the application of these herbs as an anti-aging agent is still limited and there is a lack of information in this area.

Folklore has used herbs or medicinal plants for treating various conditions such as gout, high blood pressure, diabetes, diarrhoea and also to delay aging [36]. The World Health Organization reported that most of diseases have been treated with traditional herbs or medicinal plants and this has been practiced by 65–80% of the world’s population [37]. Some of these plants have been shown to have very potent radical scavenging activity [38]. The natural compounds that are believed to contribute to these therapeutic effects are derived from a group of secondary metabolites, especially the polyphenols. Plants naturally synthesize polyphenols provide a huge category of chemical compounds which possess antioxidant, anti-carcinogenic, anti-obesity, anti-diabetic, anti-inflammatory and other functional properties. Polyphenols could also defend against oxidative stress [39,40]. These secondary metabolites have substantial variety in their structures, which contribute to the flavor, color and sensorial attributes of the plants. Polyphenols interact with the cells at a cellular level, mainly through direct contact with receptors or enzymes that are associated with signal transduction. Modification of redox status of the cells may result in this interaction and also trigger a chain of redox-dependent responses. The effects of antioxidant and prooxidant of polyphenols have been widely defined, resulting in different effects on cell physiological routes. Cell survival may improve if polyphenols act as antioxidants. In contrast, they may cause apoptosis and inhibit tumour evolution when they act as prooxidants. However, polyphenols protective effects may expand well beyond only overcoming oxidative stress [41]. Accumulating studies have reported that polyphenols such as quercetin, epicatechin gallate, epigallocatechin gallate and gallic acid could inhibit the activity of proteolytic enzymes and tyrosinase in vitro, by performing as complex or precipitating agents [42,43,44,45]. Therefore, herbs or medicinal plants which demonstrated antioxidant characteristics and inhibitory effects against aging enzymes might be applied as antioxidants and also as anti-aging agents. In view of this, metabolite profiling and correlations with radical scavenging activity and anti-aging properties of these plants should be performed.

Metabolic variation between organisms can be evaluated through metabolomics, which is considered to be a versatile tool nowadays [46,47]. The variation can be studied using specific analytical methods together with multivariate data analysis. For instance, the most frequently used analytical instrument for plant metabolomics study is the nuclear magnetic resonance (NMR) spectroscopy [48]. The NMR spectroscopy could offer a simple structural analysis achieved from the signals and coupling constants of metabolites (primary and secondary) that are present in the organisms, including crude extracts [49]. Moreover, the metabolite signal intensity relative to molar concentration can provide useful evidence about the quality and quantity of the identified metabolites [49]. NMR is the best option for fingerprinting and discrimination of herbs due to this evaluation reflecting the number of metabolites that had been identified and not the signals assigned [49]. A large quantity of data obtained from NMR analyses were managed by principle component analysis (PCA) and partial least squares (PLS), which are components of multivariate data analysis. These useful tools are also used to detect possible markers in the samples [48].

In view of the limited application and lack of information of these selected herbs as anti-aging agents, in this study the leaves of six selected herbs, which are *C. longa, O. javanica*, *V. negundo*, *P. indica*, *C. caudatus* and *P. minus,* were analysed using ^1^H-NMR spectroscopy for metabolites profiling and correlation with radical scavenging activity and anti-aging properties. It was hypothesized that these herbs would reveal significant biological activities (radical scavenging activity and anti-aging properties) and significant correlation between metabolites and anti-aging properties would be obtained. These herbs were selected based on their use for generations in local traditional cuisine and their being eaten raw as salad during the folk ages.

## 2. Results and Discussion

### 2.1. Radical Scavenging Activity of Herbs

Free radicals have been associated as the cause of numerous human diseases comprising diabetes mellitus, atherosclerosis, inflammatory lesions, ischaemic heart disease, metabolic disorders, different immunosuppressive diseases, neuromuscular degenerative conditions and in the aging process [50,51,52]. The potential mechanism of phenolic compounds as an antioxidant defense system is believed to be due to their direct scavenging of the free radicals that are involved in aging and other chronic diseases [53,54,55]. Basically, the radical scavenging activity of natural compounds in plants is measured in order to rank and recommending the best plant materials for consumption [56].

In this study, the potential of radical scavenging activity of six herbs against 2,2-diphenyl-1-picrylhydrazyl (DPPH), 2,2-azinobis(3-ethyl-benzothiazoline-6-sulfonic acid) (ABTS) and oxygen radical absorbance capacity (ORAC) (peroxyl) radicals was examined and is shown in Figure 1. The results of the study revealed that the radical scavenging activity of the studied herbs ranged from 36.12 to 411.61 μg/mL for the IC_50_ of the DPPH assay, 70.27 to 854.98 mg Trolox equivalent antioxidant capacity (TEAC)/g sample for the ABTS assay and 1972.31 to 6639.01 mM Trolox/g sample for the ORAC assay. The DPPH radical scavenging activity findings indicated that the leaves of *P. minus* showed a significantly (*p* < 0.05) higher radical scavenging effect followed by *C.caudatus*, *P.indica*, *V. negundo*, *O.e javanica* and *C. longa* leaves (Figure 1A). The leaves of *P. minus* extracted with 60% ethanol showed exceptionally high radical scavenging activity that was close to that of Trolox. This finding is in agreement with Huda-Faujan et al. [57] who stated that *P. minus* extract showed greater antioxidant activity when compared to other common herbs such as *C. caudatus*, *O. javanica*, *Centella asiatica* and *Murraya koenigii* leaves, and the antioxidant activity was comparable to that of synthetic antioxidant butylhydroxytoluene (BHT). Other studies have also proved that *P. minus* leaves demonstrated high antioxidant activity [30,58,59].

In this study, ABTS radical cation decolourization assay showed quite similar result obtained in the DPPH radical scavenging activity (Figure 1B). *P. minus* leaves showed highest Trolox equivalent antioxidant capacity (TEAC), followed by *C. caudatus*, *P. indica*, *V. negundo*, *C. longa* and *O. javanica*. However, the antioxidant value for *C. longa* leaves was not significantly different (*p* > 0.05) from *V. negundo* and *O. javanica*. This finding has been anticipated since ABTS and DPPH radical scavenging activities share a similar mechanism of reaction in the antioxidant analysis in which one or more electrons are transferred to reduce the target compounds. In the ABTS assay, the results were compared by using a Trolox equivalent. According to the TEAC, *P. minus* leaves extract was a potent radical scavenging agent and could be further studied for other functional properties.

Figure 1 also demonstrates the scavenging activity of herbs extracts on peroxyl radicals in terms of ORAC value (mM Trolox/g sample). Among the herb extracts tested, the scavenging activity of *V. negundo* and *C. caudatus* against AAPH-generated peroxyl radicals was significantly (*p* < 0.05) higher, followed by *P. minus* and *P. indica* and the lowest radical scavenging potential was exhibited by *O. javanica* and *C. longa* (Figure 1C). This finding was slightly different when compared with the other two radical scavenging assays (DPPH and ABTS), which revealed that *P. minus* had the highest free radical scavenging effect. In vitro antioxidant potential can be determined according to two categories of classes, which are hydrogen atom transfer (HAT) based assays and electron atom transfer (ET) based assays. Assays based on ET include DPPH and ABTS radical scavenging capacity assays, the total phenolic assay by Folin–Ciocalteu reagent, superoxide dismutase (SOD) assay and the ferric ion reducing antioxidant power (FRAP) assay whereas an example of HAT-based assays is the ORAC assay which scavenges peroxyl radicals [60]. Therefore, it can be speculated that the mechanism of action in the radical scavenging effect of *P.minus* was strongly dependent on ET-based assays, which mainly scavenge reactive oxygen species other than peroxyl radicals. Nevertheles, all assays suggested that *P. minus* and *C. caudatus* had high antioxidant activity, comparable to positive control, quercetin. Therefore, the application of these herbs as anti-aging agents should be investigated.

### 2.2. Elastase Inhibitory Activity

Elastase is an enzyme from the class of the protease (peptidase) class that breaks down elastin.Elastin is located in connective tissue and is a protein accountable for the elasticity of the skin and lungs and is catalyzed by elastase enzyme [61,62,63]. Previous studies have proven that deterioration of elastin by intracellular elastase increases as we age or due to repeated exposure to UV-radiation, which subsequently leads to skin aging [62,63,64].

The present study was designed to determine the effect of selected herb extracts as anti-aging agents. Figure 2A shows the inhibition of elastase enzyme by the herb extracts. Elastase enzymes incubated with the herb extracts was significantly (*p* < 0.05) inhibited the enzyme by more than 40%. The results of the study revealed that the elastase inhibitory activity of the herbs tested ranged from 43.47 to 57.91% inhibition. At a concentration of 100 µg/mL for each herb extracts, *P.minus* demonstrated significantly (*p* < 0.05) higher elastase inhibition at 57.61 ± 0.95%, even higher than that of quercetin at the same concentration with an inhibition at 52.94 ± 1.18%. *C. caudatus* also showed comparable anti-elastase activity (51.34 ± 0.21%) to quercetin, followed by *P. indica*, *V. negundo*, *C. longa* and *O. javanica*. Similar findings were also observed by other studies on *Persicaria* species (synonyms *Polygonum* spp) [43,65,66]. Kim et al. [44] also reported that the *Persicaria* spp demonstrated highest elastase inhibition among 60 plants studied with inhibition against elastase activity was more than 50%. The inhibition of the elastase enzyme activity of *P. minus* obtained in this study is in agreement with previous findings.

Plant extracts have been widely investigated for aging enzymes inhibitory activities and often found to have anti-elastase activity, for example, anti-elastase activity was determined from polyphenols isolated from persimmon (*Diospyros kaki*) leaves [67] and rosemary (*Rosmarinus officinalis*) extracts [68]. Phenolic compounds such as catechin, epicatechin, resveratrol and procyanidin B2 in grape pomace as well as plant phenolics [69,70], and flavonoids such as quercetin, kaempferol and myricetin as individual compounds [71], have also been found to have a potent anti-elastase activity. A similar study on 23 plant extracts revealed that white tea had the highest anti-elastase activities, which contained very high phenolic content [72]. Therefore, phenolic compounds especially flavonoids that could present in *P. minus* might have contributed to its high anti-elastase activity.

### 2.3. Collagenase Inhibitory Activity

A collagenase enzyme is a metalloproteinase that breaks the peptide bonds in collagen. This enzyme also is able to cleave other molecules found in cells, such as elastin, fibronectin, gelatin, aggrecan and laminin, besides collagen [73]. Results of the study exhibited that the collagenase inhibitory activity of the herbs tested ranged from 56.49 ± 1.29 to 71.00 ± 1.08%. A comparable trend was noticed for the anti-collagenase activities of the studied herbs, where *P. minus* again showed significantly (*p* < 0.05) higher collagenase inhibitory activity with 71.00 ± 1.08% (Figure 2B). Interestingly, the inhibitory effect of *P. minus* was higher than that of quercetin at the same concentration (65.95 ± 0.56%). *C. caudatus* and *P. indica* on the other hand, showed comparable anti-collagenase activity to quercetin at 66.05 ± 0.74% and 66.29 ± 1.68, respectively. The inhibition of the collagenase enzyme decreased in the order of *P. minus* > *P. indica* > *C. caudatus* > *C. longa* > *O. javanica* > *V. negundo.* However, the anti-collagenase activity was found to be not significantly different (*p* > *0.05*) between *P. indica* and *C. caudatus*, *C. longa* and *O. javanica*. A study on anti-collagenase activity has reported that green tea catechins such as epigallocatechin gallate (EGCG) could be a potent collagenase inhibitor. Similarly, in the present study, EGCG demonstrated particularly good anti-collagenase activity at 12.5 ug/mL. According to Kim et al. [74], tea extract, which contains catechins that are recognised as metal chelators, may bind to the Zn^2+^ ion within the collagenase enzyme and therefore keep it from combining with the substrate. This mechanism is due to collagenase being a zinc-containing metalloproteinase enzyme.

Collagen which is the most abundant protein in the extracellular matrix and as a major component of the skin, is responsible for skin strength and elasticity, and for maintaining its flexibility [75]. Hence, the compounds that demonstrate anti-collagenase activity may have valuable effects in general to retain healthy skin by inhibiting dermal matrix degradation via delaying the process of collagen breakdown and subsequently the wrinkling process. In this study, the inhibition effect of *P. minus* extract on collagenase activity might have occurred via several mechanisms. Since *P. minus* is a herb and most of the herbs are known to be rich with polyphenols, the hydroxyl groups (OH) of polyphenols might have connected with the backbone or other functional side group chain of collagenase. Apart from that, the hydrophobic interaction between the benzene ring of polyphenol and collagenase might have possibly changed the enzyme conformation, thus resulting in a non-functionality of the enzyme [76]. The third mechanism might involve the zinc ion active site at collagenase as stated above. Bigg et al. [77] reported that the active site of the collagenase enzyme contains a structural zinc ion which vitally function in aiding interaction with an inhibitor. In addition to the polyphenols, flavonoids, which are a subgroup of polyphenols, also could chelate zinc metal by its 3-hydroxyflavon structure [78]. 

A study by Sin and Kim [79] revealed that flavonols were very effective as collagenase inhibitors. According to the authors, quercetin and kaempferol had a higher inhibitory effect on collagenase activity as compared with flavones, isoflavones and flavanones, where the latter was the most inefficient. Therefore, the flavonols group of compounds such as quercetin, kaempferol, myricetin and galangin which might be present in *P. minus* are believed to provide the high anti-collagenase activity of this herb. On the other hand, natural elastase and collagenase inhibitors from the plant are required for a new source of anti-aging agents, in order to avoid the skin became aged or wrinkled. Therefore, the present finding suggests that there is the possibility for the same components in *P. minus* extract that may have an inhibitory effect against elastase and collagenase enzymes.

### 2.4. ^1^H-NMR Spectra of Herbs and Metabolites Identification

Representative ^1^H-NMR spectra of the studied herbs are presented in Figure 3A. Visual comparison of the ^1^H-NMR spectra of the six herbs exhibited some differences, especially for varying classes of metabolites. There was a noticeable distinction among the herbs ^1^H-NMR spectra at the aliphatic (δ 0.5–3.00), carbohydrate (δ 3.00–5.50) and in particular, the aromatic region (δ 5.50–9.00). At the aromatic region, the ^1^H-NMR spectra of *P. minus* showed marked differences and demonstrated even stronger signals as compared to the other herbs (Figure 3B). This suggested that the leaves of *P. minus* might consist of a higher concentration of aromatic metabolites as compared to the other herbs.

Further analysis on the ^1^H-NMR spectra revealed a total of 29 major metabolites were recognized from the leaf extracts, as shown in Table 1. The identification of primary metabolites was provided by the software (Chenomx database) and secondary metabolites were identified from online database (Human Metabolome Database [HMDB]; http://www.hmdb.ca/) and also by comparing with the data from the literature. The metabolites were nine primary metabolites (n = 9), phenolic acids (n = 5), flavonoids (n = 14) and ascorbic acid (n = 1). The primary metabolites identified included amino acids (alanine and valine), fatty acid and d-limonene where their chemical shifts discovered in the aliphatic region (δ 0.50–3.00). Primary metabolites consisting of carbohydrates (α-glucose, β-glucose, fructose, sucrose), choline and ascorbic acid were detected in the chemical shift region of δ 3.00–5.50. Secondary metabolites of the aromatic region (δ 5.50–9.00) were identified as quercetin, quercetin-3-*O*-rhamnoside [47,80], quercetin-3-*O*-glucoside [80], quercetin-3-*O*-glucuronide [81], quercetin-3-*O*-arabinofuranoside [47,80], rutin, myricetin derivatives, catechin, epicatechin, isorhamnetin, astragalin, apigenin, chlorogenic acid, gallic acid, coumaric acid, fumaric acid, formic acid, 3-methylxanthine and serotonin. Various studies have shown the presence of phenolic compounds and secondary metabolites, in particular flavonoids in P. minus leaves [82,83,84], as found also in this study.

### 2.5. Classification of Herb Extracts by Principal Component Analysis

The metabolite variation between the leaves of the tested herbs was further analysed using multivariate data analysis (MVDA). Principal component analysis (PCA), a pattern recognition method, is an unsupervised MVDA that delivers major understanding regarding the association between the samples. The PCA score plots showed the separation of herbs into clusters, whereas loading plots highlighted the metabolites that provide the separation [85,86]. The PCA model exhibited good fitness (R^2^X = 0.997) and high predictability (Q^2^ = 0.993) where the variation between R^2^X_(cum)_ and Q^2^_(cum)_ was less than 0.3, thus indicating that each of the herb extracts equally and evenly contributed to the observed group separation. This observation was in alignment with Wheelock et al. [87].

The principal component analysis score plot demonstrated that the selected herbs were separated into two clusters without any remarkable outliers as shown in Figure 4A. Principal component (PC)1 exhibited the greatest sample variation, then followed by PC2. PC1 and PC2 contributed to percentage of variance at 29.7% and 24.9%, respectively. Therefore, a total variance of about 54.6% was described by these PCs. Results from the loading column plot revealed the metabolites accountable for the separation of the herbs into positive side of PC1 (*V. negundo* and *C. longa*) and negative side of PC1 (*P. minus, P. indica, O. javanica* and *C. caudatus*) (Figure 4B). 

Data from the loading column plot of PC1 discovered that the phenolic compounds, mainly the flavonoid group and phenolic acids, were accountable for the separation of the selected herbs. Quercetin (1), quercetin-3-*O*-rhamnoside (2), quercetin-3-*O*-glucoside (3), quercetin-3-*O*-glucuronide (4), quercetin-3-*O*-arabinofuranoside (5), rutin (6), myricetin derivatives (7), catechin (8), epicatechin (9), isorhamnetin (10), astragalin (11), chlorogenic acid (12), gallic acid (13), coumaric acid (14), ascorbic acid (15), formic acid (21), fumaric acid (22), 3-methylxanthine (26) and apigenin (28) contents were mostly higher in *P. minus, P. indica, C. caudatus* and *O. javanica* as they were situated in the negative side of the plot. In contrast, *V. negundo* and *C. longa* were differentiated from the other herbs by the presence of serotonin (27) and d-limonene (29) in their extracts. 

### 2.6. Correlation between Bioactivities and the Metabolites Using Partial Least-Squares Analysis (PLS)

In order to comprehend the association between the measured bioactivities and metabolites found in the herbs tested, PLS, as a supervised MVDA methodology, was implemented to correlate the independent variables data (NMR chemical shift of the metabolites) to the data of dependent variables, which were the inhibition of the DPPH, ABTS and ORAC assays, as well as anti-elastase and anti-collagenase activities. This methodology was applied because PLS has a great achievement in order to link the tested bioactivities with metabolites, thus can provide a model for prediction [88]. Through the PLS analysis, the relationship between bioactivities such as radical scavenging activities and anti-aging properties with metabolites in the samples could be established. Hence, metabolites that were responsible as bioactive markers could then be suggested.

The biplot is a mixture of scores and loading plots resulting from the PLS analysis, as reported by Mediani et al. [80]. A partial least-squares biplot for the radical scavenging activity (Figure 5A) and anti-aging properties (Figure 5B) showed that all samples were well-separated and clustered without notable outliers. The PC1 separated the leaves of *P. minus, C. caudatus* and *P. indica* from *V. negundo, O. javanica* and *C. longa.* Based on the radical scavenging activities and anti-aging properties biplots, the model presented good fitness (R^2^Y) values of 0.988 and 0.966, respectively. Meanwhile, the predictability (Q^2^) for radical scavenging activities and anti-aging properties were at 0.984 and 0.951, respectively. 

As shown from PLS biplot of radical scavenging activities (Figure 5A), the bioactivities (DPPH, ABTS and ORAC assay) were directed to the positive side of the biplot, which were the most active areas and closest to *P. minus, P. indica* and *C. caudatus*. In contrast, *V. negundo, O. javanica* and *C. longa* were directed to the negative sides of the biplot, which were considered as the least active areas and were further from the DPPH, ABTS and ORAC assays, and showed negative correlation with bioactivities. In this situation, *P. minus, P. indica* and *C. caudatus*, were clustered apart from the least active herbs, indicating that these herbs exhibited a stronger radical scavenging effect, thereby suggesting that these herbs extracts might have contained higher levels of phenolic compounds. Among the three most active herbs, *P. minus* was found to be more strongly correlated with the DPPH and ABTS assay, followed by the ORAC assay.

This finding was in agreement with the in vitro results of radical scavenging activities that had been performed. The results demonstrated that *P. minus* exhibited a potent free radical scavenging effect as compared to the other herbs. The results thus revealed that *P. minus* is the most active herb for the reactive oxygen species eradicator. Significant secondary metabolites that contributed to the radical scavenging activity of *P. minus* were identified as quercetin, quercetin-3-*O*-rhamnoside, catechin, isorhamnetin, astragalin and apigenin. All these metabolites were located closer to the *P. minus* and also to the DPPH and ABTS radical scavenging assays. A previous study by Mediani et al. [80] also discovered that a freeze dried sample of plant showed a higher amount of α-glucose, β-glucose, catechin and chlorogenic acid, which might have contributed to the potent DPPH radical scavenging capacity of the herb. However, the high radical scavenging effect of *P. minus* could also be contributed by the unidentified metabolites in the extract.

Similar findings were also found for the anti-aging properties. Figure 5B presents the biplot obtained from the PLS of anti-aging properties. The bioactivities (anti-elastase and anti-collagenase activities) were projected on the positive side of the biplot, which was the most active area and were closer to *P. minus, P. indica* and *C. caudatus*. In contrast, *V. negundo, O. javanica* and *C. longa* were directed to the negative side of the biplot, which was considered as the least active area and were further away from the anti-elastase and anti-collagenase activities. This showed a negative or weaker correlation to the bioactivities. In this situation, *P. minus, P. indica* and *C. caudatus*, were clustered apart from the least active herbs, indicating that these herb extracts exhibited greater elastase and collagenase inhibition. Among the three most active herbs, *P. minus* again was found to have strong correlation with these anti-aging properties as compared to the other herbs.

This finding was in agreement with the in vitro results of anti-aging properties that had been performed earlier. The results revealed that *P. minus* was the most active herb for inhibiting elastase and collagenase enzymes. Significant secondary metabolites that contributed to the anti-aging properties of *P. minus* were identified as quercetin, quercetin-3-*O*-rhamnoside, myricetin derivatives, catechin, isorhamnetin, astragalin and apigenin. Other metabolites such as α-glucose, β-glucose, fumaric acid and fatty acid might also be responsible in the bioactivities. All of these metabolites were located closer to *P. minus* and to the anti-elastase and anti-collagenase activities.

The discrimination of these selected herbs was in agreement with the high radical scavenging activity samples especially for *P. minus, P. indica* and *C. caudatus,* which were expected to be discriminated from the others due to their high concentration of secondary metabolites, especially flavonoid. The presence of these phenolic compounds is also believed to contribute to the greater elastase and collagenase inhibitory activities of these herbs [67,68,69,70,71,72,73,74,75,76,77,78,79,89,90,91,92].

The contribution of bioactive compounds of plant extracts towards free radical scavenging ability and anti-aging activity has been documented by various studies [66,72,75,79,92,93,94,95,96,97,98,99,100,101]. In addition, metabolites such as flavonoids (quercetin, kaempferol, myricetin, epicatechin and catechin) and other phenols such as resveratrol and procyanidin B2, have been proven to significantly inhibit elastase and collagenase [69,70,71,79].

In the present study, the metabolite signals for variable importance in the projection (VIP) values were identified and reviewed to obtain the most significant metabolites that were correlated with the tested bioactivities. This was done to increase the integrity of the results presented here, which examined the discriminative potential of the identified metabolites. Generally, the metabolites signal with VIP > 0.5 was taken into consideration to be significant for discrimination [102]. In this study, all of the metabolites contributing to the radical scavenging activities and anti-aging properties could be classified as significant discriminating metabolites since their VIP values were above than 1.0 (Table 2). The two PLS biplots model was validated using 100 random permutations, to confirm the validity (R^2^) and predictive (Q^2^) abilities of the original model with several models, comparative to the goodness of fit. The R^2^ illustrated that the model fitness was significant and explained the grade of Y variables in the model, whereas Q^2^ offerred the model predictive quality similar as reported by Eriksson et al. [103].

Generally, when the values of R^2^ and Q^2^ are nearing to 1, it reflects improved presentation of the model in relation to goodness of fit and predictive quality [80]. In this study, R^2^ and Q^2^ values for both of the PLS models fell in the range of 0.951–0.988, which indicated outstanding goodness of fit (R^2^Y(cum) > 0.8) and superior predictive ability (Q^2^(cum) > 0.8). Results revealed that all the Y-axis intercepts of R^2^ and Q^2^ for the assays in radical scavenging activities and anti-aging properties were within the limits of R^2^ < 0.3 and Q^2^ < 0.05. The R^2^ and Q^2^ intercepts values were in the range of 0.0367–0.0872 and −0.444 to −0.492, respectively, suggesting that both PLS models were valid and did not show over fit. Therefore, these two PLS models could be categorized as good performance models and these findings increased the reliability of the models.

### 2.7. Relative Quantification of Secondary Metabolites

The relative quantification of some of the secondary metabolites that had been identified from the selected herbs is shown in Figure 6. These metabolites were found mostly higher in the most active herbs such as *P. minus,* were located at the positive side of the biplots and were closer to almost all the bioactivities tested. These results revealed that secondary metabolites especially from the flavonoid group of compounds that are present in high amounts in *P. minus,* might have contributed to the free radical eradicator and anti-aging properties of this herb.

When comparing to the different structures of the flavonoid that could have contributed to their effectiveness, it was noted that the hydroxylation pattern in the B-ring might be one of the important factors for the inhibition effect of the metabolites on aging enzymes activity [80]. Sim et al. [104] also revealed that at both the protein and mRNA level, the inhibitory effect of these flavonoids became powerful with a growing number of OH groups in the B-ring, and they examined the structure-activity association of some flavonoids on MMP-1 gene expression in UV-A irradiated human dermal fibroblasts.

## 3. Materials and Methods

### 3.1. Chemicals and Reagents

Deuterated methanol-*d*_4_ (CH_3_OH-*d*_4_), non-deuterated KH_2_PO_4_, sodium deuterium oxide (NaOD), trimethylsilyl propionic acid-*d*_4_ sodium salt (TSP), ethanol and methanol were supplied by Merck Millipore International (Darmstadt, Germany). Quercetin, phosphate buffer, 2,2-diphenyl-1-picrylhydrazyl (DPPH), 2,2-azinobis(3-ethyl-benzothiazoline-6-sulfonic acid) [ABTS], trolox, potassium persulfate, 2,2’-azobis(2-amidinopropane)[AAPH], epigallocatechin gallate (EGCG), HEPES buffer, elastase enzymes, *N*-methoxysuccinyl-Ala-Ala-Pro-Chloro, *N*-Methoxysuccinyl-Ala-Ala-Pro-Val-*p*-nitroanilide and deuterium oxide (D_2_O) were supplied by Sigma (Aldrich, Germany).

### 3.2. Plant Material and Sampling

Fresh leaves of *O. javanica*, *P. minus* and *C. longa* were collected from Felda Sungai Koyan Satu, Raub, Pahang. *V. negundo* leaves were obtained from Institute of Bioscience, Universiti Putra Malaysia. Fresh leaves of *P. indica* were harvested at University Agriculture Park, Universiti Putra Malaysia and fresh leaves *C. caudatus* leaves were collected at Agricultural Institute, Serdang, Selangor. Voucher specimen of these herbs were placed at the herbarium, Institute of Bioscience, Universiti Putra Malaysia and each specimen was validated by the botanist. All the leaves were harvested consistently in the morning, on sunny days to ensure reliability of metabolites content. The plot at open field for each herb was separated into six parts and each sample was collected from each segment as six replicates. 

### 3.3. Sample Preparation

Once harvested, the fresh leaves were washed under running tap water to remove all residues, dried with laboratory tissue paper and instantly frozen with liquid nitrogen to stop all the enzymatic reaction and preserved the metabolites prior to lyophilization. The samples were then dried in a LABCONCO (Kansas City, MO, USA) freeze dryer until consistent weight and the moisture content reached below 10%. All the dried samples were ground using laboratory blender to fine powder and sieved using laboratory test sieve (ENDECOTTS LTD. London, England) sized 300 μm to obtain uniform size. The powdered samples were vacuum-packed in an aluminium packaging to protect from light exposure and humidity and stored at −80 °C prior to analyses.

### 3.4. Extraction

The extraction procedure described by Mediani et al. [105] was followed with some modification. Briefly 10 g of each powdered sample were immersed in 100 mL 60% ethanol in an amber conical flask and sonicated for 1 h using an ultrasonic bath sonicator (WiseClean, model WUC-D10H, Seoul, Korea) under controlled temperature (below 40 °C). The samples were filtered through Whatman filter paper no. 1 and the remains were re-extracted twice and filtered after the first extraction completed. The extracts were then pooled and concentrated using a rotary evaporator under vacuum at 40 °C. The derived viscous substances were then freeze dried using a LABCONCO freeze dryer to ensure the comprehensive elimination of water and stored at −80 °C until further use. Lastly, the dried crude extracts were diluted to the necessary concentrations for all the investigations conducted.

### 3.5. DPPH Radical Scavenging Activity

Radical scavenging activity of the samples was determined with following the technique developed by Kong et al. [106], which was developed from a modified method of Brand-Williams et al. [107], with slight modification. Briefly, 50 μL sample extracts in methanol at different concentrations (0 to 500 μg/mL) were added with 195 μL freshly prepared 0.2 mM methanolic 2,2-diphenyl-1-picrylhydrazyl (DPPH) solution and stirred. All test samples were prepared in a 96 well plate. The decolourizing process was recorded at 515 nm using a spectrophotometer (Biotek EL 800 microplate reader, Bio-tek, Winooski, VT, USA) after 60 min incubation in darkness and compared to a positive control and blank samples. The percentage of radical scavenging activity was measured according to the following equation:% inhibition = [(A_control_ − A_sample_)/A_control_] × 100
where A_control_ is the absorbance of control without plant extracts and A_sample_ is the absorbance of plant extracts.

The plant extracts or positive control concentrations that scavenged 50% of stable free radical DPPH was calculated as the IC_50_ using linear graph of radical scavenging activity percentage against plant extracts/positive control concentrations. Lower IC_50_ indicated higher antioxidant activity. All experiments were run in six replicates with Trolox and quercetin as positive controls.

### 3.6. ABTS Radical Scavenging Assay

For ABTS radical scavenging assay, the analysis was conducted following the procedure described by Arnao et al. [108] with some amendments. The stock solutions prepared were 7 mM ABTS^+^ solution and 2.45 mM potassium persulfate solution. In order to prepare the working solution, the two stock solutions were mix in the same quantities and left to react in the dark for 16 h at room temperature. Then, the working solution was diluted with distilled water to acquire an absorbance of 0.700 ± 0.005 units at 734 nm using a spectrophotometer (UV-1650PC spectrophotometer, Shimadzu, Kyoto, Japan) and known as ABTS^+^ solution. The ABTS^+^ solution was freshly prepared for every assay. This ABTS^+^ solution (900 µL) was allowed to react with 100 µL of herb extracts for 2 min. The absorbance was then read at 734 nm using the spectrophotometer. The standard curve comprising of 3.1 μg/mL to and 50 μg/mL Trolox was developed, and the results were expressed as mg Trolox Equivalent Antioxidant Capacity/g sample (mg TEAC/g sample). 

### 3.7. ORAC Radical Scavenging Assay

An ORAC assay to measure the peroxyl radical scavenging efficacy was implemented as stated by Huang et al. [109] using The FLUOstar OPTIMA microplate fluorescence reader (BMG LABTECH, Ortenberg, Germany). Each herb extract and Trolox (standard) were prepared in 75 mM phosphate buffer solution (PBS) pH 7.4. In 96-well black microplate, a total of 150 μL fluorescein (10 nM dissolved in PBS) was added followed by 25 μL of Trolox, plant extracts or PBS as a blank. These solutions were pipetted in triplicate wells. The microplate was incubated for 15 min at 37 °C and covered with a lid. The fluorescence was measured with excitation wavelength at 458 nm and emission wavelength at 520 nm. The background signal was determined by taking the measurement every 90 s.

After that, 25 μL freshly prepared 2,2’-azobis(2-amidinopropane) (AAPH, 240 mM in PBS) was introduced by on-board injectors. The decay of fluorescence was then taken up to 90 min using the same excitation and emission wavelengths. Evaluation was done for the areas under the curve for samples (fluorescence versus time) minus the area under the curve for the blank and compared to a standard curve (25–400 μM Trolox). The ORAC values related to the Trolox was calculated using the equation as follows:ORAC value = (AUC_sample_ − AUC_blank_)/(AUC_trolox_ − AUC_blank_) 
where AUC = area under the curve. The results were expressed as millimoles of Trolox equivalents (TE)/g sample.

### 3.8. Elastase Inhibition Assay

The elastase inhibitory activity was determined according to the technique described by Kraunsoe et al. [110], with minor modifications by Ndlovu et al. [75]. The sample wells contained 25 μL 0.1 M HEPES buffer (pH 7.5), 25 μL herb extract (100 μg/mL) and 25 μL elastase enzyme (1 μg/mL). The blank wells only contained 75 μL HEPES buffer and the negative control wells contained 50 μL HEPES buffer and 25 μL elastase enzyme. The positive control wells contained 25 μL HEPES buffer, 25 μL N-methoxysuccinyl-Ala-Ala-Pro-Chloro (10 μg/mL) and 25 μL of elastase enzyme. The solvent control wells contained 25 μL HEPES buffer, 25 μL of 10% methanol and 25 μL elastase enzyme. Blank controls for the herb extract (for color controls of every extract tested) contained 150 μL HEPES buffer and 25 μL of the herb extract. The micro-well plate was then incubated for 20 min at room temperature. Following this, 100 μL substrate (*N*-Methoxysuccinyl-Ala-Ala-Pro-Val-*p*-nitroanilide, 1 mM) was added. Then the plates was incubated for an additional 40 min at 25 °C. Following incubation, the absorbance was read using a spectrophotometer (Biotek EL 800 microplate reader) at 405 nm. Percentage inhibition of the herb extracts was calculated using the equation as follows:Inhibition (%) = [(A_control_ – A_test_) / A_control_] x 100
where A_control_ is the absorbance of buffer with elastase and solvent and A_test_ is the absorbance of buffer, elastase and herb extract or *N*-Methoxysuccinyl-Ala-Ala-Pro-Chloro.

### 3.9. Collagenase Inhibition Assay

The collagenase inhibition activity was conducted following the method of Van-Wart and Steinbrink [111] (1981) with modifications by Mandrone et al. [92]. The test was implemented in 50 mM TES buffer (pH 7.4 with 0.36 mM CaCl_2_). Collagenase enzyme from *Clostridium histolyticum* (ChC–EC.3.4.23.3) was prepared at the concentration of 0.8 units/mL (dissolved in TES buffer stock solution). The synthetic substrate *N*-[3-(2-furyl) acryloyl]-Leu-Gly-Pro-Ala (FALGPA) was prepared at concentration of 2 mM in TES buffer stock solution. The total volume of 150 μL for the final reaction mixture contained 46.3 μL TES buffer, 60 μL FALGPA (0.8 mM FALGPA final concentration), 18.7 μL collagenase enzyme (0.1 units/mL final concentration) and 25 μL herb extracts (100 μg/mL). Herb extracts and enzyme in TES buffer were incubated at room temperature for 15 min before adding the substrate to initiate the chemical reaction. Negative controls were carried out with TES buffer. After addition the substrate, the absorbance was immediately read at 340 nm using a spectrophotometer (Benchmark Plus Microplate, Bio-Rad 170-6930, Bio-Rad, Hercules, CA, USA ) in a 96 well microtiter plates and continuously measured for another 20 min. Positive control were performed using epigallocatechin gallate (EGCG) at 12.5 μg/mL. Percentage inhibition of samples was calculated using equation as follows:Inhibition (%) = [1 − (A_control_ − A_test_)/A_control_)] × 100
where A_control_ is the absorbance of TES buffer and A_test_ is the absorbance of TES buffer, collagenase enzyme and plant extract or FALGPA.

### 3.10. Metabolite Profiling Using ^1^H-NMR Measurement

The metabolite profiling using ^1^H-NMR of selected herbs was performed based on the protocol described by Kim et al. [47] with slight modifications. Crude herb extracts (25 mg) were transferred into a 2 mL microcentrifuge tube. A mixture of methanol-*d*_4_ and KH_2_PO_4_ buffer in D_2_O (pH 6.0) containing 0.1% trimethylsilypropionic acid sodium salt (TSP) were added to the herb samples with the total volume of 0.7 mL at (1:1) ratio. The microcentrifuge tubes containing herb extracts were then vortexed for 1 min at room temperature followed by ultrasonication for 15 min. Then, the mixture was centrifuged for 10 min at 5678 *g* to separate the supernatant from any unsolvable materials. Next, 0.6 mL clear supernatant was incorporated into NMR tubes and subjected to ^1^H-NMR analysis. The analysis of ^1^H-NMR was accomplished via a 500 MHz Varian INOVA NMR spectrometer (Varian Inc., Palo Alto, CA, USA) which operated at a frequency of 499.887 MHz and spectra recorded at 26 °C. Every single spectrum comprised of 64 scans with 3.53 min of acquisition time and the width of 20 ppm. The data was analyzed using Chenomx software (v. 6.2) (Chenomx Inc, Edmonton, AB, Canada) to conduct phasing and baseline correction with a consistent setting. Multivariate data analysis was conducted using SIMCA software (version 13.0, Umetrics, Umeå, Sweden).

### 3.11. Bucketing of ^1^H-NMR Spectra

Bucketing of ^1^H-NMR spectra was implemented by using Chenomx software (v. 6.2, Edmonton, AB, Canada). All spectra were binned from 0.5–10.0 ppm region with the same parameters. The parameters included a spectral width of δ 0.04 which obtained a total of 245 integrated regions for each NMR spectrum. Chemical shift for water and residual methanol-*d*_4_ at δ 4.70–4.90 and δ 3.27–3.35 respectively, were eliminated. The uniform binned data was then subjected to multivariate data analysis (MVDA).

### 3.12. Relative Quantification of Metabolites

The identified metabolites were examined for their relative quantification which was calculated based on the mean peak area of the signals after binning of ^1^H-NMR spectra.

### 3.13. Statistical Analysis

All outcomes of six replicates were expressed as the means ± standard deviation. For the measurements of radical scavenging activities, inhibition of elastase and collagenase activities, and relative quantification of metabolites, statistical analyses was conducted using Minitab 16 (Version 16, Minitab Inc., State College, PA, USA). Analyses of variance (ANOVA) was applied to examine for the significant differences between the means with *p* < 0.05 considered as significantly different. Principal component analysis (PCA) and partial least square (PLS) from the multivariate data analysis (MVDA), were implemented using SIMCA-P software (v. 13.0, Umetrics, Umeå, Sweden) using the Pareto scaling method after the binning of NMR spectra was completed. The correlation between metabolites components and IC_50_ values for DPPH radical scavenging activity were converted to 1/IC_50_ in order to acquire the same trend as the functional properties activity. 

## 4. Conclusions

The application of ^1^H-NMR analyses coupled with MVDA was successful in investigating the variation in the metabolites of the selected herbs. The PCA score plot showed distinct separation of the herbs according to their clusters. This study revealed that *P. minus* possessed the highest radical scavenging effect through DPPH and ABTS assays and exhibited the highest anti-aging activities. The two biplots of PLS analysis further validated this result, since strong association was found between the metabolites identified in *P. minus* and the bioactivities tested. The active metabolites believed to contribute to the radical scavenging activities and anti-aging properties of *P. minus* are included quercetin, quercetin-3-*O*-rhamnoside, myricetin derivatives, catechin, isorhamnetin, astragalin and apigenin. Therefore, it can be assumed that these metabolites were responsible for the potent radical scavenging effect and high anti-aging properties of *P. minus*. These metabolites were mainly from the flavonoids group, in particular flavonols. Therefore, it can be suggested that the metabolites from *P. minus* could be a promising free radical eradicator and aging enzymes inhibitor that can be used for delaying aging symptoms and for the treatment of aging-associated chronic diseases. These findings will aid to establish the potency of *P. minus* as a potential natural source of anti-aging agents and as a natural free radical eradicator.

## Figures and Tables

**Figure 1 molecules-24-03208-f001:**
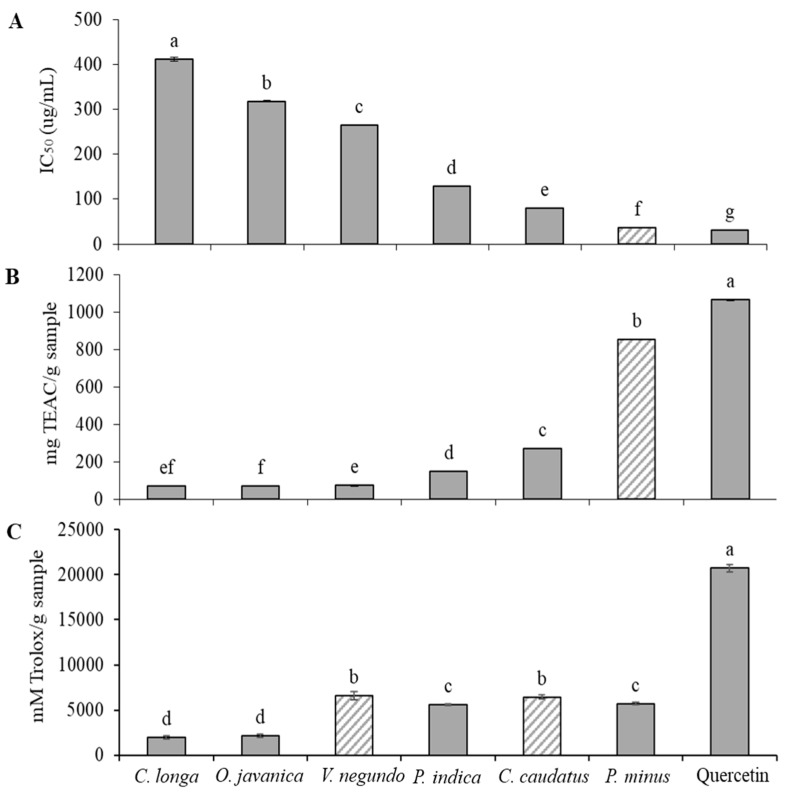
Radical scavenging activity of selected herbs leaves. **A**: 2,2-diphenyl-1-picrylhydrazyl (DPPH) assay, **B**: 2,2-azinobis(3-ethyl-benzothiazoline-6-sulfonic acid) (ABTS) assay, **C**: Oxygen radical absorbance capacity (ORAC) assay, IC_50_: extract concentration required for 50% inhibition. Means with the same letters are not significantly different (*p* > 0.05).

**Figure 2 molecules-24-03208-f002:**
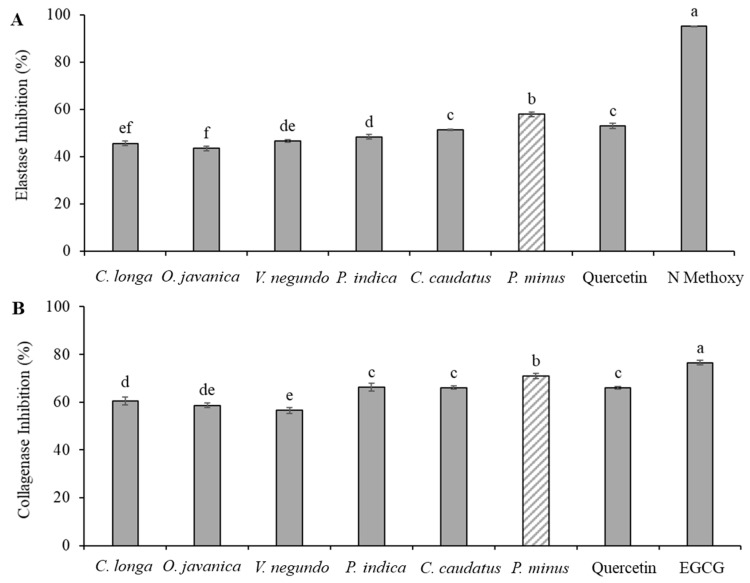
Inhibition of enzymes activity by selected herb extracts at 100 ug/mL. **A**: Elastase inhibitory activity, **B**: Collagenase inhibitory activity. Positive control *N*-Methoxysuccinyl-Ala-Ala-Pro-Chloro (*N* Methoxy) and epigallocatechin gallate (EGCG) concentration at 12.5 ug/mL. Means with the same letters are not significantly different (*p* > 0.05).

**Figure 3 molecules-24-03208-f003:**
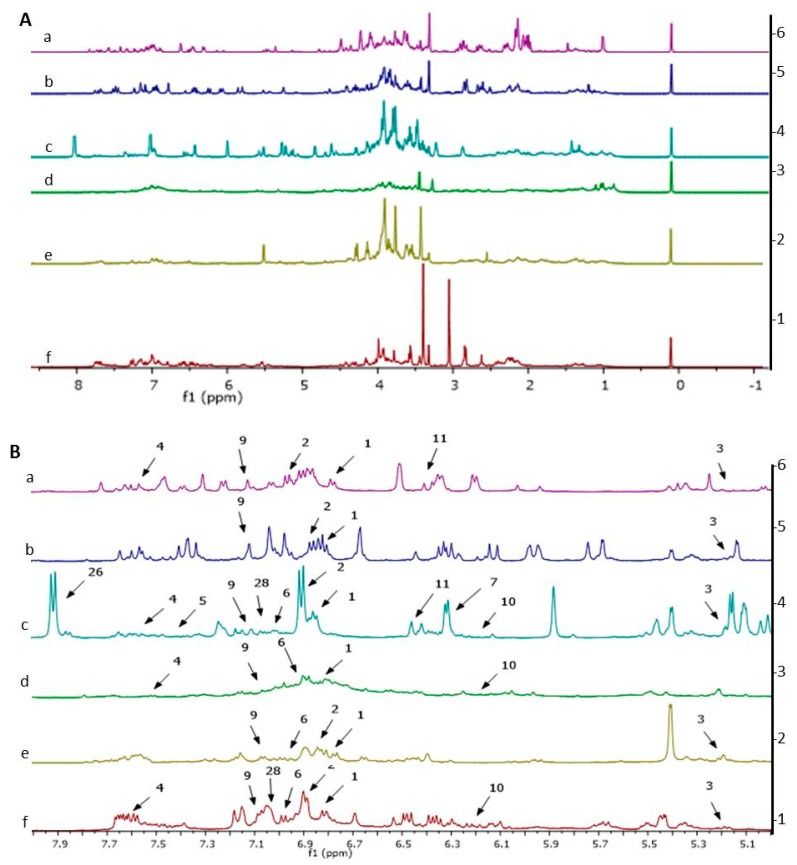
Full ^1^H-NMR spectra of selected herbs (**A**) and expanded ^1^H-NMR spectra of aromatic region from δ 5.5 to 9.0 (**B**). a: *P. indica*, b: *C. longa*, c: *P. minus*, d: *V. negundo*, e: *O. javanica*, f: *C. caudatus*. The numbering of the signals refer to the metabolites listed in Table 1.

**Figure 4 molecules-24-03208-f004:**
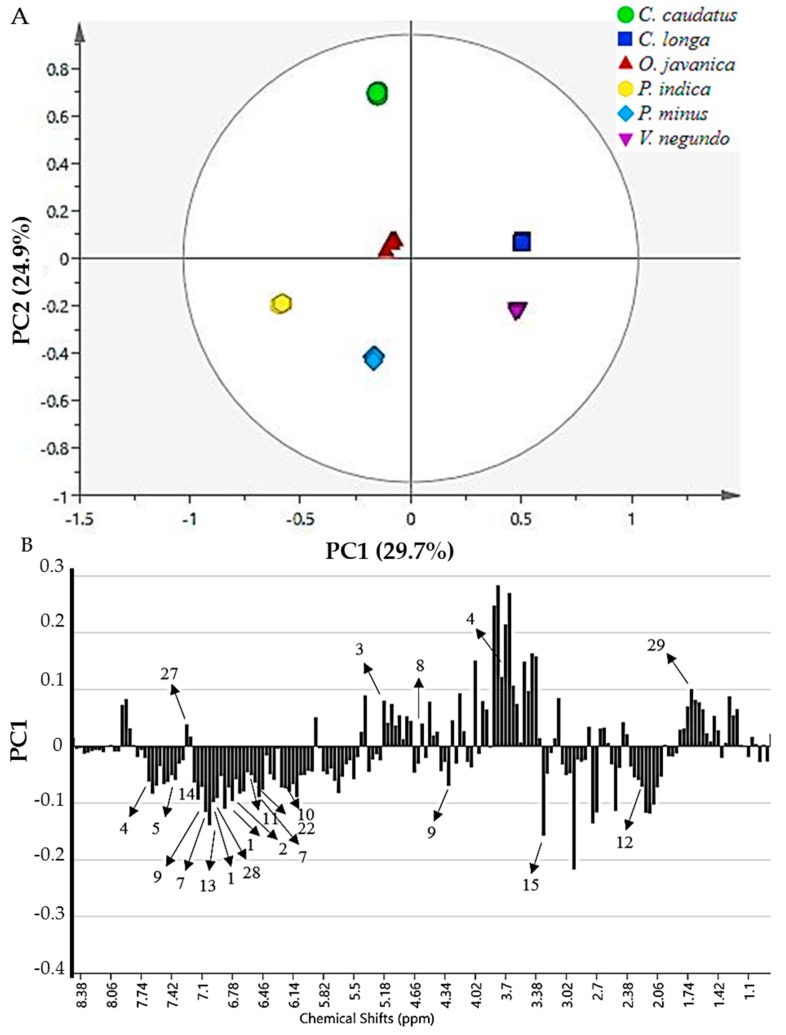
The principal component analysis (PCA) score plot (**A**) and the loading column plots (**B**) of the ^1^H-NMR data representing all the selected herbs. The numbering of the signals refers the to the metabolites in Table 1.

**Figure 5 molecules-24-03208-f005:**
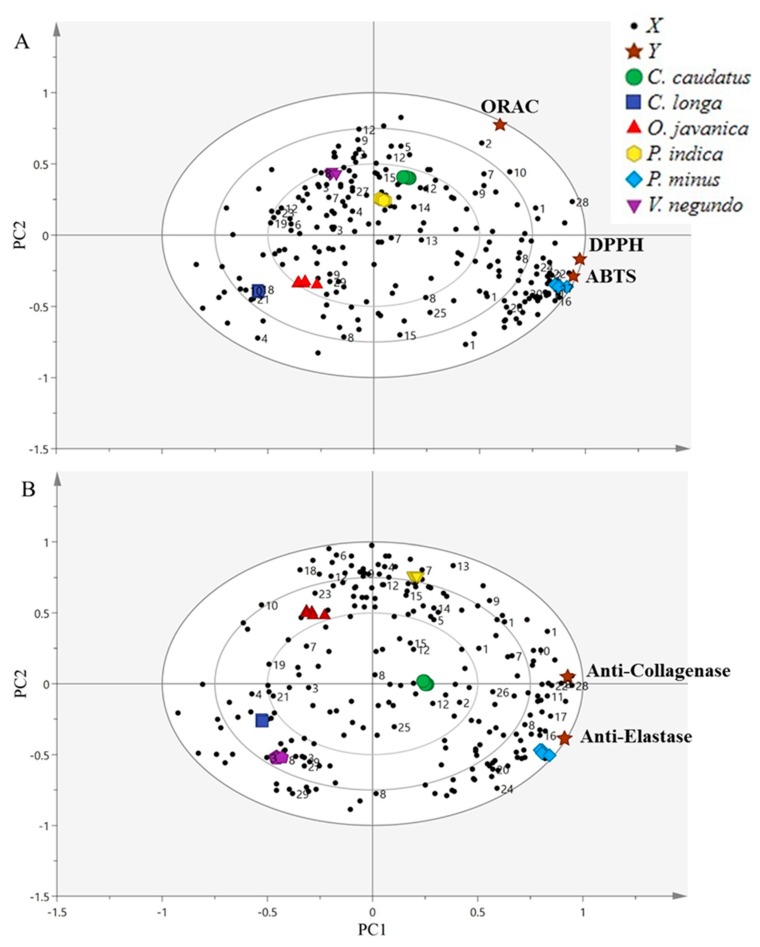
The biplots obtained from partial least squares (PLS), indicating the relationship of the metabolites variations with bioactivities tested in selected herbs. **A**: Radical scavenging activity; **B**: Anti-aging properties. The numbering of the signals refers to the metabolites in Table 1.

**Figure 6 molecules-24-03208-f006:**
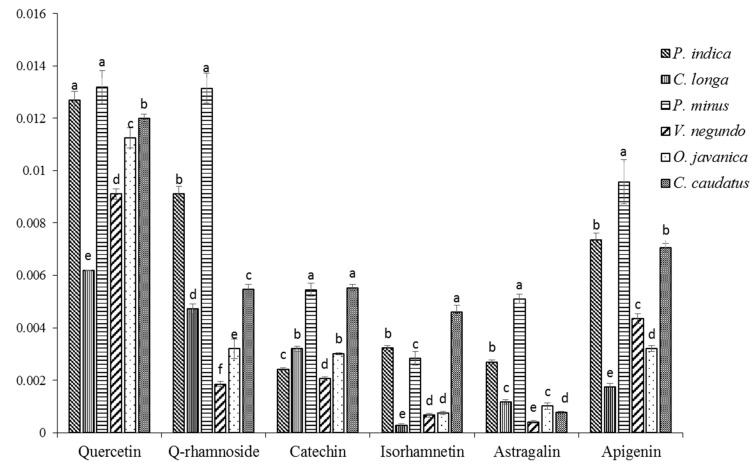
Relative quantification of identified metabolites in leaves of selected herbs based on the mean peak area of ^1^H-NMR signals. Means with the same letters are not significantly different (*p* > 0.05) within a metabolite.

**Table 1 molecules-24-03208-t001:** ^1^H-NMR characteristic signals of identified metabolites in selected herbs.

Metabolites	^1^H-NMR Characteristics Signals	Herbs					
		*Curcuma Longa*	*Oenanthe Javanica*	*Vitex Negundo*	*Pluchea Indica*	*Cosmos Caudatus*	*Persicaria Minus*
Quercetin	6.83 (d, *J* = 8.5 Hz) 6.85 (d, *J* = 8.5 Hz) 6.98 (d, *J* = 8 Hz)	+	+	+	+	+	+
Quercetin 3-*O*-rhamnoside	6.83 (d, *J* = 8.5 Hz) 6.89 (d, *J* = 8.5 Hz) 6.79 (d, *J* = 8.5 Hz) Methyl signal: 0.91 (d, *J* = 6.5 Hz)	+	+	−	+	+	+
(3) Quercetin 3-*O*-glucoside	6.86 (d, *J* = 8.5 Hz) 6.83 (d, *J* = 8.5 Hz) 5.16 (d, *J* = 7.5 Hz) 5.04 (d, *J* = 8.0 Hz) Anomeric proton glucosyl 4.97 (d, *J* = 7.5 Hz)	+	+	−	+	+	+
Quercetin 3-*O*-glucuronide	7.64 (s) 6.85 (d, *J* = 8.5 Hz) 6.87 (d, *J* = 8.0 Hz) 3.73 (d, *J* = 9.0 Hz)	−	−	+	+	+	+
(5) Quercetin 3-*O*-arabinofuranoside	7.47 (dd, *J* = 8.5 Hz, 1.7 Hz)	−	−	−	−	−	+
Rutin	6.95 (d, *J* = 8.5 Hz) 7.57 (d, *J* = 2.0 Hz) 6.92 (d, *J* = 8.5 Hz) Anomeric proton glucosyl 4.97 (d, *J* = 7.5 Hz) 4.99 (d, *J* = 7.5 Hz)	−	+	+	−	+	+
Myricetin derivatives	7.05 (s) 6.51 (d, *J* = 2.0 Hz) 6.30 (d, *J* = 2.0 Hz)	−	−	−	−	−	+
Catechin	4.59 (d, *J* = 7.5 Hz) 4.60 (d, *J* = 7.5 Hz) 4.61 (d, *J* = 7.5 Hz) 4.60 (d, *J* = 8.0 Hz) 3.93 (m) 2.83 (m) 2.84 (m) 2.56(dd, J = 16.5 Hz, 8.0 Hz)	+	+	+	+	+	+
Epicatechin	4.30 (s) 4.99 (s) 5.00 (s) 5.01 (s) 5.03 (s) 7.08 (s) 7.10 (s) 7.11 (s)	+	+	+	+	+	+
Isorhamnetin	3.84 (s) 3.85 (s) 6.21 (d, *J* = 8.0 Hz) 6.23 (d, *J* = 8.0 Hz) 6.92 (d, *J* = 8.5 Hz)	−	−	+	−	+	+
Astragalin	6.85 (d, *J* = 8.5 Hz) 6.56 (d, *J* = 2.5 Hz)	−	−	−	+	−	+
Chlorogenic acid	2.08 (m) 2.20 (m) Signal for quinic 4.04 (m) 1.88 (d, *J* = 12.0 Hz) 1.90 (d, *J* = 10.5 Hz)	−	−	−	−	+	+
(13) Gallic acid	7.03 (s)	+	+	−	−	−	+
Coumaric acid	7.17 (d, *J* = 8.0 Hz) 7.18 (d, *J* = 8.5 Hz) 7.06 (s)	−	+	−	−	+	+
Ascorbic acid	4.54 (d, *J* = 7.5 Hz) 3.29 (m)	−	−	+	−	−	+
(16) α-glucose	5.20 (d, *J* = 3.5 Hz)	−	−	+	−	+	+
(17) β-glucose	4.62 (d, *J* = 7.5 Hz)	+	+	+	−	+	+
(18) Fructose	4.20 (d, *J* = 9.0 Hz) 4.20 (d, *J* = 8.0 Hz)	+	+	+	+	−	+
(19) Sucrose	5.44 (d, *J* = 3.5 Hz) 5.42 (d, *J* = 3.5 Hz)	+	+	+	+	+	+
(20) Fatty acid	1.34 (m) 1.36 (m)	−	−	−	−	+	+
(21) Formic acid	8.48 (s)	+	−	−	−	+	−
(22) Fumaric acid	6.56 (s)	−	+	−	−	−	−
(23) Choline	3.24 (s) 3.23 (s) 3.25 (s) 3.25 (s) 3.22 (s)	+	+	+	+	+	+
(24) Alanine	1.51 (d, *J* = 7.5 Hz) 1.50 (d, *J* = 7.0 Hz) 1.49 (d, *J* = 7.0 Hz)	+	+	−	+	+	+
(25) Valine	1.08 (d, *J* = 7.0 Hz) 1.07 (d, *J* = 7.0 Hz) 1.05 (d, *J* = 7.5 Hz)	+	+	+	+	+	+
(26) 3-methylxanthine	8.02 (s)	−	−	−	−	−	+
(27) Serotonin	7.28 (s)	+	−	+	−	−	−
(28) Apigenin	6.95 (d, *J* = 8.5 Hz)	−	−	−	−	+	+
(29) d-Limonene	1.72 (s) 1.89 (m)	+	−	+	−	−	−

+: presence/strong signal of a particular metabolites in the herb extracts; −: absence/weaker signal of a particular metabolites in the herb extracts.

**Table 2 molecules-24-03208-t002:** The variable importance in the projection (VIP) values of the significant metabolites contributing to the separation and bioactivities in the biplots (PLS model).

Chemical Shift (ppm)	Metabolites	VIP Values	
		Radical Scavenging Activity Biplot	Anti-#ging Properties Biplot
6.78	Quercetin 3-*O*-rhamnoside	2.01017	2.43408
6.94	Apigenin	1.86661	1.99206
6.58	Astragalin	1.31507	1.43942
6.86	Quercetin	1.30401	1.43471
2.82	Catechin	1.07724	1.24570
6.22	Isorhamnetin	1.03878	1.23268

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
