# Peer review of "NMR-Based Metabolomics Profiling for Radical Scavenging and Anti-Aging Properties of Selected Herbs"

_molecules, 2019, doi:10.3390/molecules24173208_

Round 1

Reviewer 1 Report

Dear authors:

In my opinion the content of this paper is almost free of any major mistakes; however, it is necessary improve the spectra of V. negundo, and  O. javanica extract in order to confirm the presence of the mentioned compounds.

Author Response

Dear reviewer, thank you so much for your comments and suggestions.

We have expended the spectra and include the new version in the revised manuscript. 

Reviewer 2 Report

This manuscript presents the 1H NMR-based metabolomics profiling for radical scavenging and anti-aging properties of six selected herbs. However, the 1H NMR assignments for the compounds listed in Table 1 are not exact. For example, 1H NMR signals for quercetin (1) and quercetin 3-O-rhamnoside (2) are inconsistent with those described in the literature (Y.-C. Chang et al., J. Chi. Chem. Soc., 2000, 47, 373). Although the compounds have just two aromatic protons with a large coupling (J = ca. 8 Hz), the authors list three chemical shifts (6.83, 6.85, and 6.98 ppm for 1; 6.83, 6.89, and 6.79 ppm for 2) for each compound. Further, since the 1H NMR chemical shifts of compounds 1-6 should be almost the same in aromatic proton region (6-8 ppm), assignments of their proton signals are impossible just for crude extracts. Thus, the authors should remove the column of ‘1H-NMR characteristics signals’ from Table 1.

Author Response

Dear reviewer, thank you so much for your comments and suggestions.

For the 1H-NMR signals of quercetin (1) and quercetin 3-O-rhamnoside (2), our findings were in agreement with previous report by Mediani et al., (2012) and Khoo et al., (2015) (please refer manuscript at line 280 in text and references no. 47 and 81). In addition, the present study was also in agreement with Abdul-Rahman, (2017, PhD thesis).

Besides that, there was also variation on the method and instrument. In the present study, the instrument used was using Varian INOVA NMR spectrometer (Varian Inc., California, USA), 500 MHz with trimethylsilypropionic acid sodium salt (TSP) as an internal standard. This method was same with  Mediani et al., (2012), Khoo et al., (2015) and Abdul-Rahman, (2017, PhD thesis).

However, a paper reported by Chang et al., (2000) used Varian Gemini at 200 MHz and Varian Unity Plus (400 MHz) NMR Spectrometer for the 1H-NMR analysis with tetramethylsilane (TMS) as an internal standard.